# Photonic and Nanomechanical Modes in Acoustoplasmonic Toroidal Nanopropellers

**DOI:** 10.3390/nano14151276

**Published:** 2024-07-29

**Authors:** Beatriz Castillo López de Larrinzar, Jorge M. García, Norberto Daniel Lanzillotti-Kimura, Antonio García-Martín

**Affiliations:** 1Instituto de Micro y Nanotecnología IMN-CNM, CSIC, CEI UAM+CSIC, Isaac Newton 8, Tres Cantos, 28760 Madrid, Spain; beatriz.castillo@csic.es (B.C.L.d.L.); jm.garcia@csic.es (J.M.G.); 2Centre de Nanosciences et de Nanotechnologies, CNRS, Université Paris-Saclay, 91120 Palaiseau, France; daniel.kimura@c2n.upsaclay.fr

**Keywords:** acoustoplasmonics, nanopropeller, chirality, broken symmetries

## Abstract

Non-conventional resonances, both acoustic and photonic, are found in metallic particles with a toroidal nanopropeller geometry, which is generated by sweeping a three-lobed 2D shape along a spiral with twisting angle α. For both optical and acoustic cases, the spectral location of resonances experiences a red-shift as a function of α. We demonstrate that the optical case can be understood as a natural evolution of resonances as the spiral length of the toroidal nanopropeller increases with α, implying a huge helicity-dependent absorption cross-section. In the case of acoustic response, two red-shifting breathing modes are identified. Additionally, even a small α allows the appearance of new low-frequency resonances, whose spectral dispersion depends on a competition between the length of the generative spiral and the pitch of the toroidal nanopropeller.

## 1. Introduction

The engineering of nanoacoustic and nanophotonic structures is relevant for the conception of novel optomechanical systems, emerging quantum and sensing technologies, and for developing new telecommunications technologies [1,2,3]. The size, shape, and material composition of a nanostructure simultaneously determine its elastic and electromagnetic behavior. The development of nanofabrication techniques over the last thirty years has enabled access to nanostructures with unprecedented nanometric length scales and resolution [4,5,6,7,8].

Chirality appears in objects whose mirror images cannot be superimposed, even after three-dimensional spatial rotations. Chirality is critical in life systems, as biologically relevant molecules are predominantly chiral (proteins, amino acids, etc.) [9]. Handedness (right vs. left, dextro vs. levo) identifies two different enantiomers in chiral structures and is a crucial element determining how systems interact with their environment [10].

Thus, the presence of chirality in a system unequivocally leads to a geometrically broken symmetry. Good examples of macroscopic systems are helices, nuts, or screws. Particularly interesting is the nut–screw pair. In resemblance to what happens with a screw with the right helicity (and pitch), which can only couple with the complementary nut, circularly polarized light interacts with a chiral structure in a different manner depending on its circular polarization helicity (left or right, levo or dextro). In resonant systems, such as metallic ones that support plasmon excitations, the optical interaction with the nanostructure is drastically enhanced. Differences in the optical response for the incidence of levo or dextro light (known as the chiro-optical effect (COE)) become prominent [11,12,13]. Among the plethora of structures that can be employed for chiral applications, toroidal propeller-like structures present peculiar properties such as COE reversal [14].

Macroscopic toroidal propellers are used in aviation and maritime transport to implement less noisy propulsion systems, as they lead to a reduction of tip cavitation and fluid vortex creation since the propeller has no tips. Aquatic toroidal propellers are also improving fuel efficiency. Although the fabrication of these structures using classical machining methods is complex in the cases above, new additive manufacturing techniques drastically change this scenario. In the nanoscopic world, advanced nanofabrication tools have enabled the construction of these systems using scanning electron lithography with unprecedented resolution and precision [14,15,16].

Developing novel nanostructures possessing simultaneous optical and acoustic resonances is at the core of active fields of research, such as optomechanics, nanomechanics, and nanophononics. In this sense, most of the reported systems rely on relatively simple systems (multilayers, plasmonic antennas, dielectric resonators) [17,18,19,20,21,22], where chirality remains elusive.

In this paper, we develop a complete theoretical study of the optical and nanomechanical characteristics of a three-lobed toroidal nanopropeller (TnP) made of gold. The study is divided into two independent parts. On the one hand, we analyze the optical response, by calculating the scattering and absorption cross-sections. We demonstrate that the spectral location of optical plasmon resonances red-shifts as a function of twist angle α of the propeller. Additionally we observe an interesting interference effect for α in the range of 90 deg. to 270 deg., where the optical response is highly dependent on the spin or helicity of the incident wave. On the other hand, we obtain the nanomechanical vibrations as a result of an initial uniform expansion produced by a temperature increase of 5 K. In this case, we demonstrate that the resonance modes supported by the untwisted system also experience a red-shift as the twist angle increases. But, at variance with the optical case, the nanomechanical response presents unexpected new resonant modes at lower energies only present for non-zero twisting angles.

## 2. Materials and Methods

The geometry of the TnP used in this study is based on a three-lobed structure of identical rings, with an inner diameter of 65 nm and a wall thickness of 20 nm, as presented in Figure 1. The centers of the rings are disposed in the vertices of an equilateral triangle; as a result, the whole structure is circumscribed in a circle with a radius of 100 nm. The twisted TnP is subsequently obtained by vertically sweeping this geometry along a helical path (L) with a varying α. That sweep, together with the vertical dimension (H = 60 nm), defines the pitch of the propeller (P=2πH/α). TnP is a singularly connected structure and having a topological genus of 3.

As mentioned above, to simulate the optical and acoustic response, we assume that the TnP is made of gold. The dispersive optical parameters used are those included in the Finite-Difference Time Domain (FDTD) method Lumerical^®^ (Vancouver, BC, Canada) corresponding to those in Ref. [23]. We would like to emphasize that the combination of this software and constants is known to give a good description of the optical response, not only in the far-field region [24], but also in the near-field region [25]. In particular, it is very well suited to describing the variations of the optical resonances produced by geometrical changes, as in this work. However, in order to have a more accurate description of the spectral location of the resonances, a refinement of the optical parameters is needed, as can be found in Refs. [26,27]. Additionally, surface plasmons, with larger damping than bulk ones, can occur even in fully 3D nanosystems. This damping difference is not accounted for in our modeling. Nevertheless, the conclusions and trends regarding the resonant behavior remain valid. In order to obtain theoretical results describing a given experimental system, the use of ellipsometric parametrizations of the actual experimentally grown materials might be required [28]. The Lumerical^®^ software (V212) suite has been used to obtain the optical scattering and absorption cross-sections, and we used an impinging circularly polarized plane wave with circular polarization along the z axis, with a normalized amplitude in the whole simulation cell. The cell geometry (4 μm × 4 μm × 7 μm) ensures that perfectly absorbing boundary conditions have a negligible effect on the electromagnetic fields obtained. We used a refined mesh in the nanostructures and in the near-field region (0.3 μm × 0.3 μm × 0.9 μm) of 2 nm × 2 nm × 2 nm, dx-dy-dz, respectively, growing uniformly up to a maximum of 40 nm out of the near-field close to the simulation boundaries, so that convergence (to the best of our numerical capabilities) is attained. The total and scattered fields were then collected to give rise to the intensity color maps and cross-sections.

The Finite-Element Method (FEM) COMSOL^®^ (Stockholm, Sweden) was used to model the acoustic response produced by an initial expansion due to an increase in temperature of 5 K of the metallic nanostructure. In our simulations, the toroidal propeller was, as mentioned above, assumed to be made of gold, standing on a layer of a silicon dioxide semi-spherical substrate. The material parameters used for the FEM simulation were the ones built into the COMSOL material library^®^. The SiO_2_ substrate was an 800 nm diameter semispherical region surrounded by perfectly matching layers (PMLs) that truncate the physical domain; see, e.g., Ref. [20].

## 3. Results

### 3.1. Optical Characteristics

As can be seen, in a twisted TnP geometry, like the one depicted in Figure 1b, there is a clear handedness. It is expected that the optical response will present a helicity dependence that should depend on the twisting angle, α. To explore that possibility, the optical response of this structure was interrogated by an impinging plane wave, whose propagation vector was perpendicular in the *x*-*y* plane (i.e., in the *z* direction, from z=+∞), and its polarization was either left-circular (LCP) or right-circular (RCP) as
(1)$$|\mathrm{LCP}\rangle =\frac{|x\rangle +i|y\rangle }{\sqrt{2}};\;|\mathrm{RCP}\rangle =\frac{|x\rangle -i|y\rangle }{\sqrt{2}},$$
where |x〉 represents a linearly polarized wave along the *x* direction while traveling along the *z* direction and |y〉 is traveling in the same direction, but linearly polarized along the *y* direction.

The optical response can be described in the form of scattering and absorption cross-sections for the untwisted nanostructure as a function of the wavelength in the visible range of the optical spectrum. Similar to a metallic gold disc [24], the system presents a plasmon resonance in the range of 0.8 μm (see Figure 2a). Another smaller resonance was also observed around 0.6 μm, expected as long as the dielectric constant of Au stays in the metallic range. It is worth noting that, for this case, there is no difference between LCP and RCP, as foreseen for a structure with no handedness.

In Figure 2b,c we present the scattering and absorption cross-sections for the resonance at wavelength $\lambda_{\mathrm{res}}=0.825\;\mu$m, as a function of the twisting angle. Noticeably, in this case, there are large differences between the LCP and RCP incoming waves, signaling the presence of chirality. However, the most prominent feature is the oscillatory pattern in the optical response. In this structure, the only relevant parameter, from the point of view of optical resonances, that changes with the twisting angle is the length of the pores forming the lateral surface of the propeller. The length of the helix used to generate the TnP (see Figure 1b) can be expressed as
(2)$$L=\sqrt{{\left(\frac{\alpha D}{2}\right)}^{2}+H^{2}},$$
where α is in radians, *D* is the effective radius of the helix, and *H* is the height, as mentioned above (see Figure 1b). Notice that the effective radius of the helix, optically speaking, will have a different dimension depending on the nature of the resonance, i.e., whether the field is predominantly located in the air or in the metal.

In order to complete the optical analysis, it is apparent that the complete spectrum of the scattering and absorption cross-sections needs to be obtained. This is shown in Figure 3, where we present the scattering and absorption cross-sections as a function of the wavelength, for twisting angles below 360 degrees.

In all panels in Figure 3, we observe that the resonances at 0.825 μm and at 0.6 μm, as a function of the twisting angle, behave in the following manner: for α from 0 to c.a. 50 deg., there is very little dispersion; for values from 50 to ∼110 deg., the resonances strongly shifts to longer wavelengths, and interestingly, for twisting angles α≳ 120 deg., they disperse linearly towards high wavelengths running parallel with each other. This happens both for the absorption and the scattering cross-sections. Noticeably, for the twisted structure (α∼120 degrees), the absorption (depicted in (c) and (d)) is over two-times bigger than for the untwisted one, but the volume of the system remains essentially the same. There are obvious differences in the intensity for LCP and RCP waves, pointing out COE, but apart from the actual value of the cross-sections, the overall oscillatory phenomenology is virtually equivalent for both helicities. In fact, the plot of Equation (Equation 2) (see Appendix A) shows that it accurately describes the trend followed by the resonance peaks, even despite the fact that some may have different slopes (that would depend on a varying effective D).

Furthermore, from a simple inspection of the panels in Figure 3, there are obvious signatures of a large COE. This COE is especially relevant in the absorption cross-section, where the response of the propeller is enhanced for LCP waves with respect to RCP ones, reaching differences larger than 100%. For the scattering cross-section, the differences are not as marked and occur at shorter wavelengths than for the absorption.

### 3.2. Acoustic Characteristics

Let us now address the characterization of the acoustic response of the TnP. For that, we study the vibrations, characterized by the displacement of each point in the harmonic regime after an initial isotropic thermal expansion produced by a ΔT=5 K.

In order to find the resonant modes corresponding to the structure, we monitor the average displacement (i.e., the 3D integral of the RMS displacement of each point of the TnP normalized to its volume). The resonant modes appear as distinct peaks, as presented in Figure 4. Two natural, low-frequency, modes for an untwisted (red curve) TnP appear at 4.6 GHz and 5.5 GHz, respectively, and correspond to the breathing modes with maximum (low-frequency) and minimum (high-frequency) displacement at the point marked A in Figure 1. As soon as the structure is slightly twisted (even by just one degree), we clearly observe two different phenomena. The first one is that the resonances associated with the breathing modes shift towards lower energies (i.e., the wavelength associated with the resonances increases), in perfect correspondence with the previously presented optical behavior. The second observation, now at variance with the optical case, is that, in the 3 to 4 GHz region, two new resonances appear, which downshift in frequency as the twist angle increases. For a zero twist angle (red curve in Figure 4 and its inset), there are no resonances in this region, but even for a twist of one degree, they appear (black curve in Figure 4’s inset). The evolution with a twist angle from α = 0 to α = 15 deg. (see the inset in Figure 4) shows a further increase in intensity and a downshifting in frequency. In Figure A2 in Appendix B, we present some snap shots for the breathing modes at α=0 deg. and for low-frequency twist-enabled modes at α=15 deg.

Similar to the optical case, we present in Figure 5 the whole evolution of the resonance location for a TnP with a complete twist angle (360 deg.). As indicated in Figure 4, we can see that the lowest energy resonances for the untwisted geometry (inset (c) of Figure 5) appear for 4.6 GHz and 5.5 GHz and evolve towards lower energies as the twist angle increases. One relevant aspect that can be extracted from this figure is that, although the resonances always evolve towards a lower energy value, there is a change in the trend. Initially, the frequency decreases roughly 1/L as expected to fit with the increase of helix length (see Equation (Equation 2)). However, this trend changes at α∼120 deg. To shed some light onto the change in slope, we refer to the features depicted in Figure 5a,b, showing the geometry for twist angles of 360 degrees and 120 degrees, respectively. If the structure were made by a single isolated lobe, in one full loop (α = 360 deg.), the height of the TnP would match the pitch of the helix, and that would be the first occasion on which one lobe in the base would have another one exactly above. However, our structure is made of three lobes, and Figure 5a resembles a much more twisted structure; unsurprisingly, it appears to have three twists. In fact, as seen in Figure 5b, it is at one third of a twist (α=120 deg.) when a lobe of the base has another one above. This is then when two effects compete: the resonance tends to decrease in energy due to the growth of L against the resistance due to the upper and lower branches of the TnP.

Eventually, for a very twisted geometry, the resonance should converge to that of a disk shape with a twisted void geometry inside, so compressed that the variation of the angle should have a negligible effect.

## 4. Discussion and Conclusions

Taking into account our results, a next step would be to devote efforts to quantifying the observed mechanical/optical mode dynamics and their optomechanical coupling.

In simple optomechanical systems such as GaAs/AlAs resonators [29,30] or bar acoustoplasmonic nanoantennas [5,31], both optical and nanomechanical resonances exhibit similar behaviors when the geometric parameters are changed. In contrast, the photonic and acoustic responses of the proposed toroidal nanopropeller depend differently on the structural parameters. Thus, optimization can be performed to access novel optomechanical regimes or functionalities, including efficient transduction of complex nanophononic modes. Understanding the origin of these optical and acoustic modes is essential to actually use them in optomechanical systems, where the helicity of light can be used as a design parameter. Toward that end, we have presented a complete study of the optical and acoustic properties of a three-lobed twisted toroidal nanoscale propeller-like structure. Taking into account our results, a next step would be to devote efforts to quantifying the observed mechanical/optical mode dynamics and their optomechanical coupling. The presence of the twist in the TnP is indicated by a red-shift in both its optical and acoustic resonances. This red-shift occurs because the wavelength couples with the effective length of the generative helix, which varies with the twist angle of the TnP. Consequently, the twist angle acts as a tuning structural parameter that influences the nature, intensity (which depends on helicity), and spectral positions of the optical and acoustic resonances. Additionally, the twist present in the TnP leads to new low-frequency resonant acoustic modes that are not present in the spectrum of an equivalent untwisted structure. These new acoustic modes appearing at non-zero twisting angles enable a new avenue in the engineering of optomechanical interactions at the nanoscale. 

## Figures and Tables

**Figure 1 nanomaterials-14-01276-f001:**
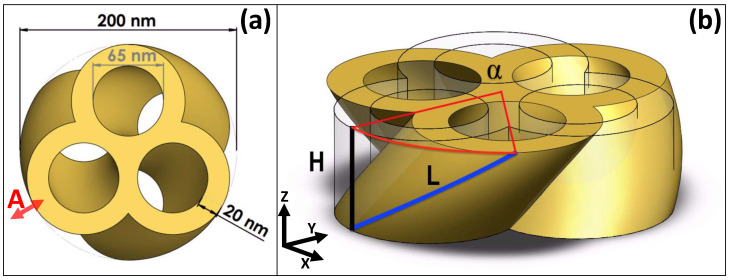
Schematic view of a α=60-degree twisted gold TnP. (**a**) Top view depicting the geometrical parameters. The nanostructure is inscribed in a circle of a 100 nm radius; the diameter of the inner hole is 65 nm; the wall thickness is 20 nm. Label A corresponds to breathing modes’ resonances (see Section 3.2). (**b**) The height is fixed to a value of H = 60 nm; α is the twisting angle. The length of the helical path (blue line) *L* increases from H (for α=0) to *L* as the twisting angle α increases (see Equation (Equation 2)).

**Figure 2 nanomaterials-14-01276-f002:**
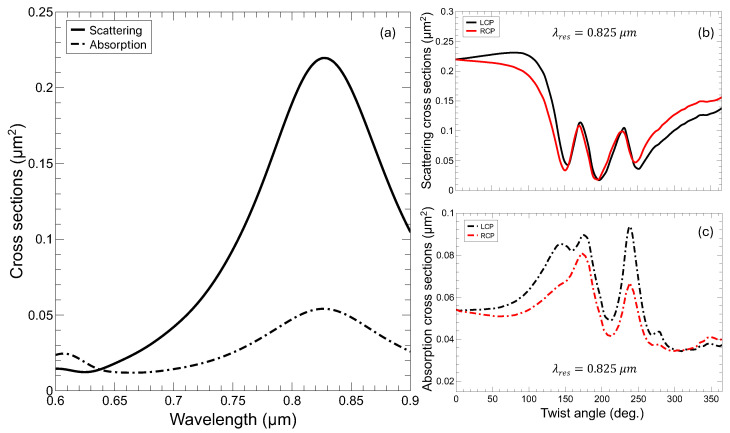
(**a**) Scattering and absorption cross-section for the untwisted TnP in Figure 1 for LCP and RCP (no differences for this geometry), clearly depicting the main plasmonic resonance at $\lambda_{\mathrm{res}}=0.825\;\mu$m and another weaker one at shorter ∼0.6μm wavelength. (**b**) Scattering and (**c**) absorption cross-sections for a fixed wavelength, $\lambda_{\mathrm{res}}$, varying the twisting angle.

**Figure 3 nanomaterials-14-01276-f003:**
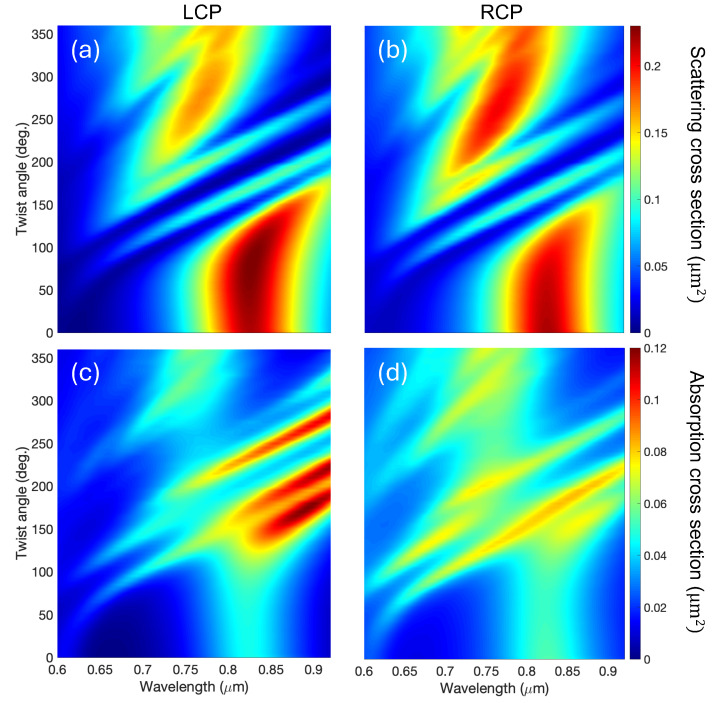
Top: scattering cross-sections for LCP (**a**) and RCP (**b**) as a function of the twist angle and of the wavelength. Bottom: absorption cross-sections for LCP (**c**) and RCP (**d**) as a function of the twist angle and of the wavelength. As can be seen, there is an almost linear dependence of the resonances for increasing twist angles. The high helicity dependence, COE, of the absorption is clearly observed in the bottom panels.

**Figure 4 nanomaterials-14-01276-f004:**
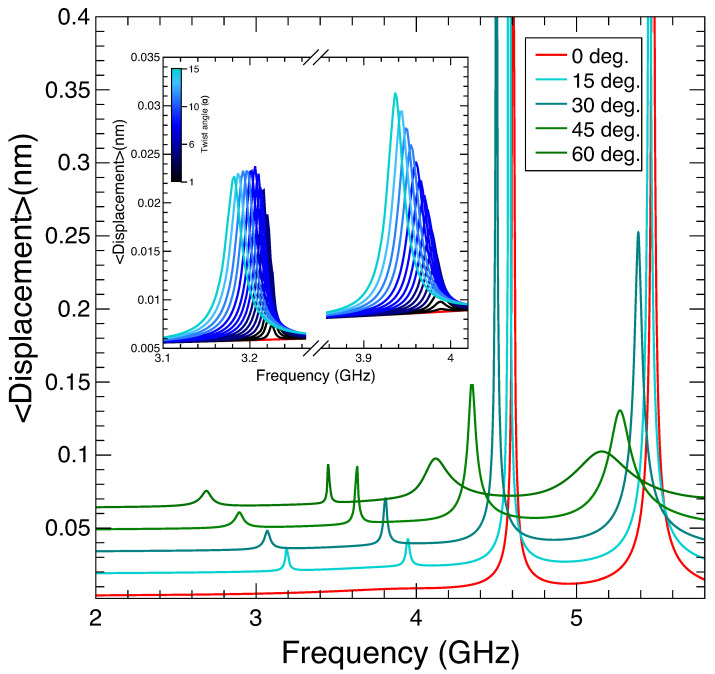
Average RMS displacement of a gold TnP as a function of the acoustic frequency for different twisting angles (0 to 60 degrees). The curves for non-zero twist angle are shifted 0.015 nm vertically for clarity. The peaks represent the nanomechanical modes of the structure. In the inset, we present the angular evolution (1 to 15 degrees) of the low-frequency modes that appear only for non-zero twist angles.

**Figure 5 nanomaterials-14-01276-f005:**
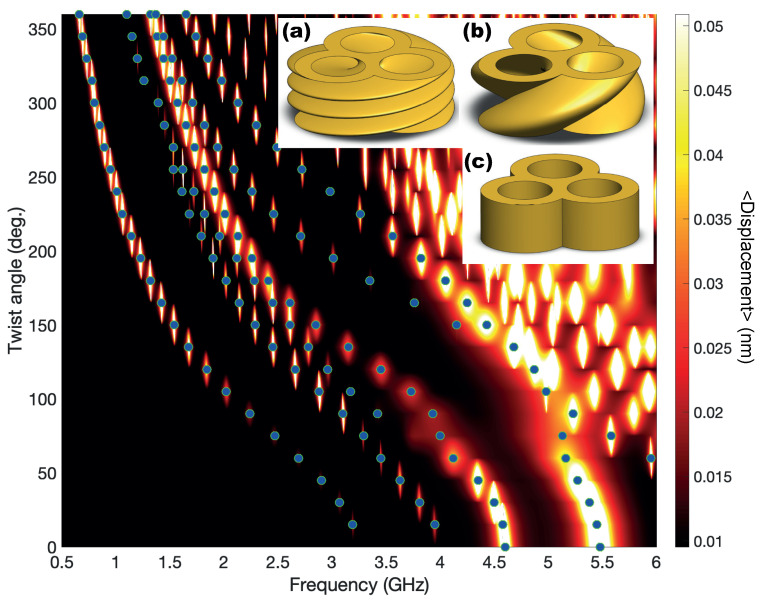
Colormap of the average RMS displacement of a gold TnP as a function of the acoustic frequency for one complete loop of the twisting angle. The intensity is saturated at 0.05 nm for clarity, and the peaks representing the five lowest nanomechanical modes of the structure are marked with dots. The insets show the geometry for a TnP with a twist angle of 360 degrees (**a**) and 120 degrees (**b**) and an untwisted TnP (**c**).

## Data Availability

The raw data supporting the conclusions of this article will be made available by the authors upon request.

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
