# Peer review of "Photonic and Nanomechanical Modes in Acoustoplasmonic Toroidal Nanopropellers"

_nanomaterials, 2024, doi:10.3390/nano14151276_

Round 1

Reviewer 1 Report

Comments and Suggestions for Authors

Title: Photonic and phononic modes in acoustoplasmonic toroidal nanopropellers

Manuscript ID:  nanomaterials-3089777

Comments to the authors:

 The authors describe investigations of the optical and acoustic response under excitation with a circular polarized plane wave of a "toroidal nanopropeller" theoretically. They used the software Lumerical and Comsol for this purpose.  The propellers made of gold are created by twisting three connected cylindrical rings with a defined height H. The article is well structured, well written and comprehensible. I have only a few minor remarks or questions:

 In general: Does H also change when twisting? If so, must this be considered in the calculations, equ.2, figure A1?

 Line 52: "... after a 5 degree kelvin thermal uniform expansion...", This is a bit confusing to read. I think you mean an expansion produced by a temperature increase of 5K. You should improve that.

 Line 77: what do you mean with “proper” polarization? Can you be more specific?

 Line 120: .., D is the radius of the helix, … , In Figure 1 you describe D as the diameter of the untwisted structure (circle) and here it is the radius der helix. That's a bit confusing. What do you mean?

 Figure 4: “< >” can be removed or is this necessary?

 Line 190: Discussion: Could you name some optomechanical systems where these structures could be used in practice?

Reviewer 2 Report

Comments and Suggestions for Authors

The paper deals with a numerical simulation study of helical nano-metallic chiral system using conventional software packets like COMSOL. The study is not, however, precisely and clearly addressed. In the Introduction we meet a mixture of notions like optical response, plasmons or phonons.

As the studied sample is a specific chiral gold nanosystem, the electro-magnetic response (plasmons) is related mainly with free electron dynamics, as usually for metals. In this place it should be emphasized that the conventional tools like COMSOL are highly simplified, i.e., they offer the numerical solution of Maxwell equations (for boundary Fresnel problem) at assumed as the prerequisite for the finite element method of solution of differential equations a material parameters – the dielectric functions both for the metal and the dielectric surroundings. In the case of metallic nanosystems this has been evidenced to be strongly misleading – spectral behavior for plasmon oscillations in metallic nanostructures experimentally observed is different in such systems in comparison to COMSOL-type simulations, thus the latter are of low value to simulate a true behavior (“Quantum Nano-Plasmoncs”, Cambridge UP 2020 and Vol. 23, No. 4 | DOI:10.1364/OE.23.004472 | OPTICS EXPRESS 4472). The reason of the discrepancy is the size-dependence of Lorentz friction in the nanoscale in comparison to bulk material. Thus, the numerical study should include different form of the metal dielectric function for a nanoparticle than that for a bulk metal (but the latter, i.e., of bulk metal, is assumed in COMSOL calculations). This must be revised or at least commented that a significant effect is neglected and the obtained results related to optical response cannot be considered as of high fidelity. The second problem  is a phonon-type response. It is unclear in the text, but phonons are primarily related in metals with ion lattice vibrations. The contribution of free electrons to phonons is small and thus a conclusion of direct mutual dependence/connection of optical response (plasmons) and of phonon-type response would be confusing as both effects are related to different subsystems of the considered structure. This must be discussed and clarified in the paper. The influence of chirality onto both effects may not be related with mutual dependence/connection but rather with common for both subsystems geometry constraints imposed onto boundaries.

Thus the paper needs the major revision to enhance its scientific background.  

Comments on the Quality of English Language

the text needs a revision/verification of used terminology

Round 2

Reviewer 2 Report

Comments and Suggestions for Authors

The Authors have revised slightly the submission along suggested recommendations in the first round of the review procedure. They have also point to point answered to posed problems and partly removed confusing earlier statements. Now they have emphasized that the optical response of plasmon type is fully independent from the mechanical phonon-type response. The Authors have substituted the term of phonon response by nanomechanical one – it is unimportant but may be used to clearly differentiate between electromagnetic response (plasmons) and mechanical one (phonons). The second objection – the limited accuracy of numerical modeling by finite element solution of Maxwell-Fresnel boundary problem  in nanoscale still needs some revision as the arguments of the Authors that they have used  Lumerical not Comsol system are rather not convincing, as both are of the similar type. The data from thin films (by ellipsometry measurements) also do not solve the problem as plasmons in such structures are still of bulk type in contrast to surface plasmons in fully 3D nanosystem as they considered (for which large damping of plasmons occurs in contrary to bulk modes and this dumping is not accounted for in simulations upon Comsol and Lumerical). Some supplementation of the related comment would be thus still of order. Some corrections are introduced without a sufficient care – as for example in lines 74, 194 (the break before citation).

Comments on the Quality of English Language

only minor linguistic proofreading would be beneficial
